# Development of Stable Amino-Pyrimidine–Curcumin Analogs: Synthesis, Equilibria in Solution, and Potential Anti-Proliferative Activity

**DOI:** 10.3390/ijms241813963

**Published:** 2023-09-11

**Authors:** Matteo Mari, Matteo Boniburini, Marianna Tosato, Luca Rigamonti, Laura Cuoghi, Silvia Belluti, Carol Imbriano, Giulia Avino, Mattia Asti, Erika Ferrari

**Affiliations:** 1Department of Chemical and Geological Sciences, University of Modena and Reggio Emilia, Via Campi 103, 41125 Modena, Italy; matteo.mari@unimore.it (M.M.); matteo.boniburini@unimore.it (M.B.); marianna.tosato@unimore.it (M.T.); luca.rigamonti@unimore.it (L.R.);; 2Radiopharmaceutical Chemistry Section, Nuclear Medicine Unit, Azienda USL-IRCCS Reggio Emilia, Via Amendola 2, 42122 Reggio Emilia, Italy; mattia.asti@ausl.re.it; 3Department of Life Sciences, University of Modena and Reggio Emilia, Via Campi 213/d, 41125 Modena, Italy; laura.cuoghi@unimore.it (L.C.); silvia.belluti@unimore.it (S.B.); carol.imbriano@unimore.it (C.I.)

**Keywords:** curcumin, pyrimidine derivatives, solution equilibria, cell proliferation, cancer cells

## Abstract

With the clear need for better cancer treatment, naturally occurring molecules represent a powerful inspiration. Recently, curcumin has attracted attention for its pleiotropic anticancer activity in vitro, especially against colorectal and prostate cancer cells. Unfortunately, these encouraging results were disappointing in vivo due to curcumin’s low stability and poor bioavailability. To overcome these issues, herein, the synthesis of eight new pyrimidine–curcumin derivatives is reported. The compounds were fully characterized (^1^H/^13^C NMR (Nuclear Magnetic Resonance), LC-MS (Liquid Chromatography-Mass Spectrometri), UV-Vis spectroscopy), particularly their acid/base behavior; overall protonation constants were estimated, and species distribution, as a function of pH, was predicted, suggesting that all the compounds are in their neutral form at pH 7.4. All the compounds were extremely stable in simulated physiological media (phosphate-buffered saline and simulated plasma). The compounds were tested in vitro (48 h incubation treatment) to assess their effect on cell viability in prostate cancer (LNCaP and PC3) and colorectal cancer (HT29 and HCT116) cell lines. Two compounds showed the same anti-proliferative activity as curcumin against HCT116 cells and improved cytotoxicity against PC3 cells.

## 1. Introduction

Cancers are the second leading cause of death worldwide after heart diseases and represent the highest impact burden on the health system, both in terms of casualties and costs for treatment and caregiving. Among them, prostate cancer (PCa) is first for the number of cases in men, while colorectal cancer (CRC) is second and third for incidence in female and male populations all over the world [1]. Generally, for localized non-metastatic prostate cancer, therapy includes local ablation through surgical or radiotherapeutic intervention with or without antihormonal treatment [2]. Castration-resistant prostate cancer remains incurable, although men suffering from this disease are living considerably longer thanks to androgen deprivation therapy combined with androgen receptor signaling inhibitors, such as abiraterone, enzalutamide, or apalutamide, with or without radiotherapy (Radium-223) and chemotherapy (docetaxel) [3,4]. CRC treatment usually requires the surgical resection of lesions, followed by chemotherapy and/or radiotherapy. Current chemotherapy includes both single-agent and multiple-agent regimens, the most commonly used drugs being fluoropyrimidine derivatives, oxaliplatin, irinotecan, and capecitabine [5]. Another approach is represented by targeted therapy that affects cancerous cells by directly inhibiting cell proliferation, differentiation, and migration [5].

In the search for new therapeutics aimed at reducing side effects and the treatment of recurrences, naturally occurring compounds have been of inspiration for their pleiotropic activities [6,7,8]. In the last two decades, curcumin [(1E,6E)-1,7-bis (4-hydroxy-3-methoxyphenyl)-1,6-heptadiene-3,5-dione], the most active phytocompound extracted from the dried rhizomes of *Curcuma longa* L., stands out, as highlighted by the exponential increase in released scientific papers, reviews, and clinical trials. Recently, many in vitro investigations have pointed out that curcumin impacts CRC cell proliferation, showing pleiotropic activity, particularly triggering G1/S arrest and apoptosis via p53 and p21 [9]. In CRC cells, curcumin reduces growth and cell invasion by altering the expression of genes controlling cell proliferation, among which EGFR is repressed as a consequence of the reduced trans-activation activity of Egr-1 [10]. Similarly, in PCa cells, curcumin affects the expression of cell-cycle- and apoptosis-related genes, such as Cyclins D1, B1 and B2, Puma, Noxa, and Bcl-2 family members [11]. Curcumin recently demonstrated interesting anti-proliferative activity against prostate cancer both in vitro and in vivo, inhibiting both androgen-sensitive and -insensitive prostate cancer cells by targeting several signaling cascades responsible for regulating cellular function [12].

Nonetheless, curcumin is not used as therapeutic yet, the main concerns being its poor absorption and fast metabolism under physiological conditions; allegedly, the keto–enol moiety is extremely reactive (weak acid/Michael acceptor/bidentate chelator) and is supposed to be responsible for the unsatisfactory results observed in vivo. To overcome this drawback and possibly enhance therapeutic efficacy, drug delivery systems have been developed, particularly hybrid materials [13,14,15], and several synthetic modifications have been studied [16,17,18,19].

Nitrogen heterocyclic moieties are frequently inserted in potential drugs as pharmacophores given the wide range of pharmacological activities, among which are anticancer properties. Particularly, pyrazoles and pyrimidines have a leading position in drug design, being interesting pharmacophores [20,21]. Various drugs containing pyrimidine moieties have been approved as potent anticancer agents, such as 5-Fluorouracil, Merbarone, and Imatinib, just to cite a few [21]. Several pyrimidine-based analogs were synthesized and evaluated for their abilities to target various protein kinase enzymes, including EGFR tyrosine kinase [22]. Osimertinib was the first FDA-approved pyrimidine-containing drug with remarkable clinical efficacy against non-small-cell lung cancer [23].

With the aim of improving the bioavailability and stability of curcumin in physiological conditions and possibly enhancing its anticancer activity, the reactive keto–enol moiety was exchanged with the pharmacophore structure of an amine-pyrimidine. Eight new derivatives (Figure 1) were synthesized and fully characterized. The effect on cellular proliferation was tested in vitro in colorectal (HCT116 and HT29) and prostate (PC3 and LNCaP) cancer cell lines.

## 2. Results

### 2.1. Synthesis

Few pyrimidine derivatives of curcuminoids have previously been obtained [19,24]]; the synthetic pathway was characterized by direct condensation of urea (or thiourea) with the di-keto moiety of curcumin [24] or by reaction in extremely acid conditions (HCl in ethanol/toluene) of a dimethyl pyrimidine derivative with a substituted benzaldehyde [24]. As for the lead compound, curcumin, all the herein-reported amino-pyrimidine derivatives (Figure 1) were obtained using Knoevenagel condensation involving the electrophilic aldehyde and the activated methyl group. The reaction does not require any specific protection step, which is different from the synthesis of curcumin, in which the methylene moiety is protected as a boron acetylacetonate complex [25]; indeed, the corresponding CH group of the pyrimidine ring (H-5) is not reactive. The mono-condensation products (MPY series) were obtained when the reaction was carried out in a one-to-one molar ratio between the aldehyde and the amino-pyrimidine, whereas to trigger the formation of the PY-symmetric product, a two-to-one molar ratio is needed. However, a mixture of MPY and PY was always observed, and chromatographic purification is mandatory to achieve a purity >95%. In the presence of a phenolic group on the benzaldehyde, the reaction was performed in acid conditions (CH_3_COONH_4_/CH_3_COOH) to deactivate the nucleophilicity of the phenolic oxygen and avoid any possible side reactions. The reaction was carried out under a basic environment (*tetra*-butylammonium hydrogen sulfide (TBAHS)/NaOH) for PY2-PY4, in these conditions the product is obtained in slightly lower yield (35–40% vs. 50–60%) but with higher purity. MPY3 is synthesized starting from MPY1 through a nucleophilic SN2 reaction, and it is used as an intermediate to form MPY4 by direct coupling with HBTU-activated Fluor-nicotinic acid. All the compounds were synthesized in the thermodynamically more stable *E* isomeric form that is easily detected using a typical ^1^H NMR spin system. As shown in Figure 2, two doublets with a characteristic scalar coupling of 16 Hz are observed for both MPY and PY derivatives (^1^H-NMR spectrum of PY1 is reported in Appendix A).

### 2.2. Acid–Base Equilibria in Solution

Pyrimidine derivatives can be considered weak bases, and the pyrimidine nitrogens are protonated in extremely acid conditions (pH < 1), while the NH_2_ group has a pK_a_ around 5 depending on the electronic effects of the substituents on the aromatic rings [26]. As a consequence, the amino-pyrimidine moiety is in the neutral form at physiological pH (7.4). Among the synthesized compounds, MPY1, MPY3, and PY1 are characterized by additional acid/base moieties, particularly a phenolic group in the meta position for MPY1 and PY1 and an additional primary aliphatic amine for MPY3. To estimate the protonation constants, UV-visible spectroscopy was preferred over potentiometry since the lower concentration range that is required allows it to work in an aqueous medium with a negligible percentage of methanol (<2% *v*/*v*). Figure 3 reports the UV-Vis pH-metric titration of MPY1 in the pH range of 2–11. In extremely acid conditions (pH 2, red spectrum), the amine group on the pyrimidine ring is fully protonated, and the molecule is positively charged (H_2_L^+^). The maximum absorbance is observed at 400 nm, with a shoulder at 500 nm. As long as the pH is increased to 7 (green spectrum), the amine dissociates, a neutral species (HL) is observed, the maximum undergoes an ipsochromic (blue) shift from 400 nm to 355 nm, and the molar extinction coefficient (ε) does not change significantly. Finally, in basic conditions (pH 11, blue spectrum) the phenolic group is deprotonated, and a negatively charged species is observed (L^−^) with λ_max_ at 406 nm. An isosbestic point is observed at 370 nm, suggesting acid/base equilibria in solution. By plotting the absorbance value vs. pH at λ_max_, a titration curve is observed (Figure 2, inset), with two equivalent points around pH = 5 and pH = 9. MPY3 (Appendix A) is characterized by a maximum absorbance at 350 nm in neutral and basic conditions that shifts to 385 nm in acid ones, suggesting the presence of a di-protonated species at pH < 5 (H_2_L^2+^) and a neutral species at pH > 9 (L). PY1 (Appendix A) is fully protonated below pH 5 (H_3_L+) and negatively charged in basic conditions (HL^−^, L^2−^). To estimate protonation constants and pK_a_ values, spectrophotometric data were elaborated using Hypspec [27], and the logβ values, together with pK_a_, are reported in Table 1. 

Overall, protonation constants are used to estimate the species distribution curves as a function of pH (Figure 4). In physiological conditions (pH 7.4), the prevailing species for all the investigated compounds is the neutral one.

### 2.3. Pharmacokinetic Stability in Physiological Conditions

Curcumin’s low bioavailability is tightly connected to its instability in physiological conditions; in order to check if the substitution of the keto–enol moiety with an amino-pyrimidine ring is effective, a pharmacokinetic study was performed, and two media were taken into account: phosphate-buffered saline (PBS) and simulated human plasma (SHP), both at 37 °C and pH = 7.4. In PBS, the precursor, 2-amino-4,6-dimethyl-pyridimine, shows maximum absorbance at 286 nm; as long as π conjugation increases, a bathochromic shift is observed for the synthesized derivatives (λ_max_~350 nm and ~390 nm for MPY and PY compounds, respectively). As shown in Figure 5, no significant decrease in absorbance is observed for the investigated compounds, suggesting stability over 95% within 8 h in PBS. The different functional groups in the aromatic rings do not affect the stability in the tested physiological media, and all the compounds are extremely stable in solutions. The only needed precaution is to avoid UV light exposure, which may activate the *trans*-to-*cis* interconversion.

### 2.4. Inhibitory Effects on Human Cancer Cells Proliferation

To determine the anti-proliferative activity of the compounds on cancer cells, we performed dose–response treatments for 48 h on androgen-sensitive and androgen-insensitive PCa cells (LNCaP and PC3, respectively), as well as CRC cell lines (HCT116 and HT29). As highlighted by the GI50 values, which represent the concentration reducing cell growth by 50% (Table 2), compound **PY1** showed similar activity compared to curcumin in all cell lines. Differently, compound **PY3** was more active than curcumin in both PCa cell lines, with GI50 values of 12.1 µM and 14.7 µM versus 35.8 µM and 23.9 µM for curcumin in LNCaP and PC3, respectively. In CRC cells, **PY3** was more potent than curcumin (GI50 = 30.6 µM) in HT29, having a GI50 of 19.7 µM, but not in HCT116 cells. 

To further investigate the effects of **PY1** and **PY3** on cell growth, we analyzed the cell cycle progression of LNCaP cells treated at GI50 doses for 48 h (Figure 6 and Appendix A). Cytofluorimetric analysis of propidium-iodide-stained cells showed an increase in apoptotic cells upon administration of curcumin (from 0.9% in control cells to 7.2% in treated cells) and a weak decrease in S-phase cells (from 12.5% to 9.7%). Compared to curcumin, **PY1** induced a similar but enhanced effect on the cell cycle, with apoptosis and S-phase being 9.6% and 6.4%, respectively. Interestingly, **PY3** showed a different anti-proliferative activity that is mainly caused by the accumulation of the G2/M population, which rises from 18.4% (DMSO) to 54.6%. This effect was similarly observed in PC3 and HCT116 cells, with G2/M cells increasing from 14.6% to 62% in PC3 and from 17% to 33% in HCT116 cells (Appendix A).

## 3. Discussion

The eight new pyrimidine curcumin derivatives were obtained by exploring both basic and acid-catalyzed reactions with satisfactory yields. Only phenolic derivatives required acid conditions to avoid phenolate reactivity as a nucleophilic moiety. Similar to curcumin synthesis [25], Knoevenagel condensation solely affords the formation of the *E* isomer, as demonstrated by the ^1^H-NMR analysis that evidences a ^3^J_H,H_ of 16 Hz previously observed for curcumin [28]. Both PY and MPY compounds were stable in the simulated physiological conditions, suggesting the key role of the keto–enol moiety in triggering the high instability of curcumin and curcuminoids. The keto–enol moiety undergoes tautomeric equilibrium, and different conformers were observed in solution [29], especially for asymmetric curcuminoids [30], providing the formation of a multitude of species in solution. Similar improvements in pharmacokinetic profiles were previously observed by the exchange of the keto–enol moiety with a pyrazole ring [31,32,33], suggesting that stiffening the core of curcumin while still maintaining π conjugations could be a good strategy in designing new derivatives. In addition, the keto–enol moiety is a weak acid characterized by a pK_a_ value of 8.56 in curcumin [28] that could be reactive as a bidentate chelating site toward biologically occurring metal ions such as iron, copper, magnesium, and calcium, to cite a few. The introduction of the amino-pyrimidine moiety avoids the kinds of interactions that might reduce the efficacy in vivo, particularly because the amino group is deactivated by the pyrimidine ring, hence its nucleophilicity and binding ability are extremely poor. The amino-pyrimidine is a weak base, characterized by an overall protonation constant of ~5 for all the synthesized derivatives, suggesting that the amino group is in its neutral form in physiological conditions. As shown in species distribution curves (Figure 4), even the compounds with more than one acid/base equilibrium are all in the neutral form.

The reduction in π conjugations in MPY compounds is responsible for their extremely weak inhibition of cancer cell proliferation, as shown by the high values of GI50 reported in Table 2. These compounds, different from Curcumin, resulted in more activity in vitro in reducing cell proliferation of prostate cancer cell lines rather than colorectal ones. Among the PY series, **PY1** and **PY3** showed the most interesting results. **PY1**, directly homologous of curcumin, was demonstrated to have similar anti-proliferative activity in AR-sensitive and AR-insensitive PCa cells, with GI50 being about 24 µM in both LNCaP and PC3 cells. The effect of **PY1** on the cell cycle progression of LNCaP cells is more robust than curcumin, and the accumulation in the G2/M population is accompanied by subG1 events, which are presumably due to apoptotic cells. As for CRC, **PY1** shows double values of GI50 in HT29than HCT116cells, which is similar to what was observed for curcumin. These two cell lines show a similar doubling time but differ in the status of the p53 gene, which is wt (wild-type) in HCT116 and mutated in HT29 cells. The anti-proliferative effect of curcumin and its derivatives are mediated by p53; therefore, we can speculate that **PY1** may have a similar p53-dependent mechanism of action [9,16,34]. **PY3** differs from **PY1** in the substitution of the phenolic groups with methoxy ones. These structural modifications were previously investigated for the curcumin analog bis-dimethoxycurcumin (DiMC), the anticancer activities of which were recently reviewed [35]. **PY3** was shown to be more active in reducing HT29 cell proliferation (IG50 19.7 µM) than curcumin (30.6 µM) and DiMC (43.4 µM) [36]. While DiMC induces G0/G1 phase arrest in CRC [37], **PY3** triggers an evident G2/M block of the cell cycle in both HT29 and HCT116 cells. **PY3** was even more effective against PCa cell lines than curcumin (12.1 µM and 14.7 µM vs. 35.8 µM and 23.9 µM, respectively, for LNCaP and PC3). Curcumin is able to induce apoptosis in PCa cells through an AR-independent pathway via NFkB [38], and DiMC significantly increases downstream apoptotic markers compared to the lead compound [39]. Once again, the similar GI50 between LNCaP and PC3 suggests that PY3 acts through AR-independent mechanisms. 

Overall, the effects induced by **PY3** on the cell proliferation of both CRC and PCa cells highlight its efficacy as an anti-cancer molecule and, potentially, its feasibility as an alternative to curcumin in further pre-clinical investigations due to its improved stability in physiological conditions, which also allows easier quantification of the compound in pharmacodynamics studies in vivo. The higher solubility of these compounds in comparison to curcumin in physiological media, together with their improved stability, may account for their potential oral administration.

## 4. Materials and Methods

All the chemicals and solvents were purchased with the highest purity grade available and used without further purification unless otherwise specified. pH measurements were carried out using a calibrated pH meter (Mettler-Toledo). Liquid chromatography/mass spectrometry (LC/MS) was performed on an Agilent 6300 Ion Trap LC/MS system equipped with an electrospray ionization (ESI) interface. Elemental analysis was performed on a Thermo Scientific™ FLASH 2000 CHNS Analyzer (Waltham, MA, USA). UV–visible spectra were recorded with a JASCO V-770 UV/Vis/NIR spectrophotometer at 298 K in a 250–600 nm spectral range employing quartz cells (1 cm optical path). Nuclear magnetic resonance (NMR) spectra were recorded on a Bruker Biospin FT-NMR AVANCE III HD (600 MHz) spectrometer equipped with a CryoProbe BBO H&F 5 mm in inverse detection. The nominal frequencies were 150.90 MHz for ^13^C and 600.13 MHz for ^1^H. Atom numbering of NMR data refers to Figure 1.

Phosphate-buffered saline (PBS) was prepared by dissolving NaCl (8.0 g), KCl (0.2 g), Na_2_HPO_4_ (1.44 g), and KH_2_PO_4_ (0.245 g) in 1 L of Milli-Q water and by adjusting the pH to 7.4 by small addition (1 µL) of conc. NaOH (4M). Simulated plasma was prepared following the procedure of Samiei et al. [40]. 

All the reaction intermediates were purified as specified in the following procedures, and their purity (≥95%) was checked by a combination of LC/MS, NMR, and elemental analysis.

### 4.1. Synthesis

*4-[(E)-2-(2-amino-6-methylpyrimidin-4-yl)ethenyl]-2-methoxyphenol* (**MPY1**)

A total of 1.2 mmol (150 mg) of 2-amino-4,6-dimethylpyrimidine and 1.1 mmol (170 mg) of vanillin were dissolved in acetate buffer (1.7 g of CH_3_COONH_4_ in 11.0 mL of CH_3_COOH). The mixture was stirred overnight (16 h) at 100 °C. During the reaction, color changes were observed (typically from yellow to red/orange). After cooling to r.t., the mixture was carefully and slowly neutralized by adding NaHCO_3_-saturated solution. The raw product was extracted three times with ethyl acetate (EtOAc). The organic phases were collected, washed with brine solution, and dried under MgSO_4_. After filtering off, the solvent was removed under reduced pressure. The raw product was purified by flash column chromatography (silica, gradient (*v*/*v*) petroleum ether (EtPet): ethyl acetate (EtOAc) 100:0 → 0:100, then up to 5% of MeOH).

Yellow/orange powder, 58% Yield. LC/MS (ESI): [M + H]^+^ 258.4 *m/z*. Elemental analysis for C_14_H_15_N_3_O_2_: *calc*. C (65.35%), H (5.88%), N (16.33%), *expt*. C (65.24%), H (5.95%), N (16.24%). ^1^H NMR (δ (ppm) DMSO-*d_6_*): 2.23 (H-7, s, 3H), 3. 84 (OCH_3_, s, 3H), 6.35 ppm (NH_2_, s, 2H), 6.55 (H-5, s, 1H), 6.80 (H-8, d, 1H), 6.84 (H-15, d, 1H), 7.03 (H-14, dd, 1H), 7.23 (H-11, d, 1H), 7.60 (H-9, d, 1H).


*4-[(E)-2-(3,4-dimethoxyphenyl)ethenyl]-6-methylpyrimidin-2-amine* (**MPY2**)


MPY2 was obtained with the same synthesis reported for MPY1, adding 3,4-dimethoxy-benzaldeheide (1.1 mmol; 180 mg) instead of vanillin. Yellow powder, 68% Yield. LC/MS (ESI): 242.1 *m/z* [M + H]^+^. Elemental analysis for C_14_H_15_N_3_O: *calc*. C (69.69%), H (6.27%), N (17.41%), *expt*. C(69.57%) H(6.35%) N(17.44%). ^1^H NMR (δ (ppm) DMSO-*d_6_*): 2.20 (H-7, s, 3H), 3. 79 (OCH_3_, s, 3H), 6.39 ppm (NH_2_, s, 2H), 6.56 (H-5, s, 1H), 6.85 (H-8, d, 1H), 6.97 (H-12/14, d, 2H), 7.58 ppm (H-11/15, dd, 2H), 7.65 (H-9, d, 1H).


*4-{(E)-2-[4-(3-aminopropoxy)-3-methoxyphenyl]ethenyl}-6-methylpyrimidin-2-amine* (**MPY3**)


MPY1 (0.392 mmol, 100 mg) was dissolved in 1 mL of dimethylformamide (DMF) together with K_2_CO_3_/KI (0.1 g/0.2 g), and *tert*-butyl (3-bromopropyl)carbamate (0.563 mmol (134 mg)/1 mL DMF) was then slowly added to the mixture. The mixture was heated up to 80 °C for 18 h under magnetic stirring. After cooling to r.t., the solvent was removed under reduced pressure. The sticky product was dissolved in Milli-Q water and extracted three times with EtOAc. The organic phases were collected and washed with brine, dried over MgSO_4_, filtered off, and the solvent was removed under reduced pressure. The raw product was purified with flash column chromatography (silica, gradient (*v*/*v*) EtPet:EtOAc 100:0 → 0:100, then up to 5% of MeOH). Brown powder, 69% Yield (raw).

The Boc-protected intermediate was dissolved in 0.5 mL of DCM and 0.5 mL of TFA, and the mixture was left for 16 h under magnetic stirring at r.t. Eventually, the organic solvent was removed under reduced pressure. The crude product was suspended in water, the pH was increased to 9 with a basic solution (saturated Na_2_CO_3_), and it was extracted with EtOAc three times. The organic phases were collected and washed with brine, dried over MgSO_4_, the solid was filtered off, and the solvent was removed under reduced pressure.

Brown powder, 82% Yield. LC/MS (ESI): 317.6 *m/z* [M + H]^+^. Elemental analysis for C_17_H_24_N_4_O_2_: *calc*. C (64.53%), H (7.65%), N (17.71%); *expt*. C (64.45%), H (7.71%), N (17.60). ^1^H NMR (δ (ppm) DMSO-*d_6_*)): 1.79 (H-18, m, 2H), 2.24 (H-7, s, 3H), 2.68 (H-19, m, 2H) 3.83 (H-16, s, 3H), 4.05 (H-17, t, 2H), 6.37 (N-H_2_, broad s, 2H), 6.57 (H-5, s, 1H), 6.90 (H-8, d, 1H), 6.98 (H-11, d, 1H), 7.12 (H-14, dd, 1H), 7.26 (H-11, d, 1H), 7.63 (H-9, d, 1H).


*N-({4-[(E)-2-(2-amino-6-methylpyrimidin-4-yl)ethenyl]phenoxy}propyl)-6-fluoropyridine-3-carboxamide* (**MPY4**)


MPY3 (0.65 mmol, 205 mg) was dissolved in 2 mL of DMF, and 6-fluoropyridine-3-carboxylic acid (0.7 mmol, 100 mg) was added together with DIPEA (1 mmol, 130 mg) and HBTU (0.8 mmol, 300 mg). The mixture was kept under magnetic stirring at r.t. overnight (18 h). The solvent was removed under reduced pressure, and the sticky raw product was dissolved in Milli-Q water and extracted three times with EtOAc. The organic phases were collected and washed with brine, dried over MgSO_4_, filtered off, and the solvent removed under reduced pressure. The raw product was purified with flash column chromatography (silica, gradient (*v*/*v*) EtOAc:acetone 100:0 → 0:100, then up to 5% of MeOH). 

Dark-yellow powder, 41% Yield. LC/MS (ESI): 438.5 *m/z* [M + H]^+^. Elemental analysis for C_20_H_18_FN_5_O_2_: *calc*. C (63.32%), H (4.78%), N (18.46%), *expt*. C (63.21%), H (4.89%), N (18.36%). ^1^H NMR (δ (ppm) DMSO-*d_6_*): 2.19 (H-18, m, 2H), 2.35 (H-7, s, 3H), 3.46 (H-19, dd, 2H) 3.82 (OCH_3_, s, 3H), 4.10 (H-17, t, 2H), 6.40 (NH_2_, broad s, 2H), 6.58 (H-5, s, 1H), 6.91 (H-9, d, 1H), 7.00 (H-14, d, 1H), 7.13 (H-15, dd, 1H), 7.28 (H-10, d, 1H), 7.31 (H-24, d, 1H), 7.64 (H-8, d, 1H), 8.38 (H-22, m, 1H), 8.70 (H-25, d, 1H), 8.75 (NHCO, t, 1H).


*4,4′-{(2-aminopyrimidine-4,6-diyl)di[(E)ethene-2,1-diyl]}bis(2-methoxyphenol) (***PY1**)


The synthetic procedure was the same as for MPY1 with double the amount of vanillin (2.5 mmol (312 mg) instead of 1.2 mmol).

Orange powder, 51% Yield. LC/MS (ESI): 392.2 *m/z* [M + H]^+^. Elemental analysis for C_22_H_21_N_3_O_4_: *calc*. C (74.39%), H (6.50%), N (10.84%); *expt*. C (74.35%), H (6.61%), N (10.60). ^1^H NMR (δ (ppm) DMSO-*d_6_*): 3.82 (OCH_3_, s, 6H), 6.55 (H1, s, 1H), 6.34 (NH_2_, broad s, 2H), 6.80 (H-11, d, 2H), 6.90 (H-5, d, 2H), 7.05 ppm (H-12, dd, 2H), 7.25 (H-8, d, 2H), 7.65 (H-6, d, 2H).


*4,6-bis[(E)-2-phenylethenyl]pyrimidin-2-amine* (**PY2**)


A total of 1.2 mmol (150 mg) of 2-amino-4,6-dimethylpyrimidine and 0.1 mmol (34 mg) of tetrabutylammonium hydrogen sulfate (TBAHS) were dissolved in 3 mL of NaOH solution (5 molL^−1^) under stirring at 50 °C. After complete dissolution, 2.5 mmol (265 mg) of benzaldehyde was added, and the temperature was increased to 100 °C. The reaction was kept under continuous stirring for 8 h, then cooled down to r.t. A raw solid was filtered off and recrystallized from ethanol. 

Light-yellow powder, 70% Yield. LC/MS (ESI): 300.1 *m/z* [M + H]^+^. Elemental analysis for C_20_H_17_N_3_: *calc*. C (80.24%) H (5.72%) N (14.04%); *expt*. C (80.16%), H (5.83%), N (14.15). ^1^H NMR (δ (ppm) DMSO-*d_6_*): 6.93 (H-3, s, 1H), 7.10 (H-5, *d*, J = 16 Hz, 2H), 7.76 (H-6, d, J = 16 Hz, 2H), 7.67 (H-8/12, m, 4H), 7.44 (H-9/11, m, 4H), 7.38 (H-10, m, 2H), 6.53 (NH_2_, s broad, 2H). 


*4,6-bis[(E)-2-(3,4-dimethoxyphenyl)ethenyl]pyrimidin-2-amine (***PY3**)


The synthesis was carried out with the same procedure as PY2, using 3,4-dimethoxybenzaldehyde (2.5 mmol; 415 mg) instead of benzaldehyde. 

Yellow powder, 35% Yield. LC/MS (ESI): 420.2 *m/z* [M + H]^+^. Elemental analysis for C_24_H_25_N_3_O_4_: *calc*. C (68.72%) H (6.01%) N (10.02%); *expt*. C (68.66%), H (6.10%), N (10.15). ^1^H NMR (δ (ppm) DMSO-*d_6_*): 6.85 (H-3, *s*, 1H), 6.97 (H-5, *d*, 2H,), 7.70 (H-6, *d*, 2H), 7.28 (H-8, *dd*, 2H), 7.00 (H-11, *dd*, 2H), 7.18 (H-12, *dd*, 2H), 3.84 (OCH_3_, *s*,6H), 3.80 (OCH_3_, *s*,6H), 6.39 (NH_2_, *s broad*, 2H).


*4,6-bis[(E)-2-(3-methoxyphenyl)ethenyl]pyrimidin-2-amine (***PY4**)


The synthesis of **PY4** was carried out with the same procedure as PY2, using 3-methoxybenzaldehyde (2.5 mmol; 340 mg) instead of benzaldehyde. 

Yellow powder, 42% Yield. LC/MS (ESI): 360.2 *m/z* [M + H]^+^. Elemental analysis for C_24_H_25_N_3_O_4_: *calc*. C (73.52%) H (5.89%) N (11.69%); *expt*. C (73.45%), H (5.94%), N (11.75). ^1^H NMR (δ (ppm) DMSO-*d_6_*): 6.84 (H-3, *s*, 1H), 6.93 (H-5, d, 2H), 7.69 (H-6, d, 2H), 7.61 (H-8, dd, 4H), 6.70 (H-9, dd, 4H), 3.81 (OCH_3_, *s*,6H), 6.42 (NH_2_, *s* broad, 2H).

### 4.2. Kinetic Stability of Ligands in Physiological Conditions

The chemical stability at 37 °C in darkness was evaluated using UV−Vis spectroscopy as a change in absorbance in a 200−600 nm range over a period of 8 h. Then, 50 µM solutions of each compound were prepared in 0.1 M phosphate-buffered solution (PBS) and simulated plasma fluid (SPF) [41] at pH 7.4. Spectra were recorded every 5 min during the first hour and every 30 min the following ones.

### 4.3. Acid/Base Character

To perform spectrophotometric titrations, ligands (denoted generally as L in the following) were dissolved in methanol to give a mother solution (2.50 mM) that was then diluted in water to a final volume of 25 mL in order to obtain a 25/50 μM concentration, depending on the compound, in order to have a maximum absorbance in a range of 0.4–1. The pH (initial value ~5) was varied by adding small amounts (1 μL) of concentrated NaOH/HCl (4 M) in order to investigate the spectral behavior of the 2–11 pH range. In these conditions, the volume variations were negligible. A constant ionic strength (NaNO_3_, 0.1 M) was maintained in all the experiments. The overall protonation constants (β_qr_) are defined by the following equations:qL^l−^ + rH^+^ ⇌ [L_q_H_r_]^(r − ql)^(1)
(2)$$\beta_{\mathrm{qr}}=\frac{\left[L_{q}H_{r}{}^{\left(r\;-\;\mathrm{ql}\right)}\right]}{[L^{-l}{]}^{q}\;\cdot \;[H^{+}{]}^{r}}$$
where L is the ligand in the completely dissociated form and H is a proton. β_qr_ values were refined from spectrophotometric data using least-squares calculations in HypSpec [27]. The results of least-squares calculations include the standard deviations and correlation coefficients of the refined parameters. The quantities were obtained by performing error propagation calculations from the experimental errors on the parameters. The stability constant refinement furnishes least-squares estimates of the standard deviation, σ, of the stability constant, β. The error on logβ is calculated as follows: σ(logβ) = [log(β + σ) − log(β − σ)]/2, as previously reported [42].

### 4.4. Cell Lines

The human colon cell lines, HT29 (ATCC Cat# HTB-38) and HCT116 (ATCC Cat# CCL-247), were grown in DMEM High-Glucose Medium (Biowest, Nuaillé, France), while the human prostate cell lines, PC3 (ATCC Cat# CRL-1435) and LNCaP (ATCC Cat# CRL-1740), were grown in Ham’s F12 (Biowest, Nuaillé, France) and RPMI 1640 (Biowest, Nuaillé, France), respectively. All media were supplemented with 2 mM glutamine, 100 IU/mL penicillin, 100 μg/mL streptomycin, and 10% fetal bovine serum (FBS, Gibco, Fisher Scientific Italia, Segrate (MI), Italia)). The cells were grown at 37 °C in a humified 5% CO_2_ atmosphere. 

### 4.5. Cell Viability Assay

HCT116, HT29, PC3, or LNCaP cells were seeded into a 96-well plate at a density of 5000 cells/well. Cells were treated with eight different concentrations of pyrimidine derivates and curcumin for 48 h, at concentrations starting at 30 μM with 2-fold serial dilutions for curcumin and starting at 100 μM with 3-fold serial dilutions for pyrimidine derivates. Cell viability was measured using Presto Blue cell viability reagent (#A13261, Thermo Fisher Scientific, Waltham, MA, USA), according to the manufacturer’s protocol. The concentration at which cellular growth is inhibited by 50% (GI50) was determined.

### 4.6. Cell Cycle Analysis

Cells were seeded into a 24-well plate at a density of 35,000 cells/well and harvested after 48 h of treatment at GI50 concentrations. Cell cycle analysis of cells stained with propidium iodide solution (propidium iodide 25 µg/mL, Na-Citrate 3.4 mM, NaCl 9.65 mM, NP-40 0.03%) was performed using an Attune Next cytofluorimeter (Thermo Fisher Scientific, Waltham, MA, USA). DMSO was used as the control. Statistical analysis was performed with GraphPad PRISM 6 software (GraphPad Prism), using two-way ANOVA as specified in the figure legends. Graph represents means ± standard errors of the mean (SEM), n = 3. Data were considered to be statistically significant if *p* <  0.05 (*), *p* <  0.01 (**), *p* <  0.001(***), and *p* <  0.0001 (****).

## Figures and Tables

**Figure 1 ijms-24-13963-f001:**
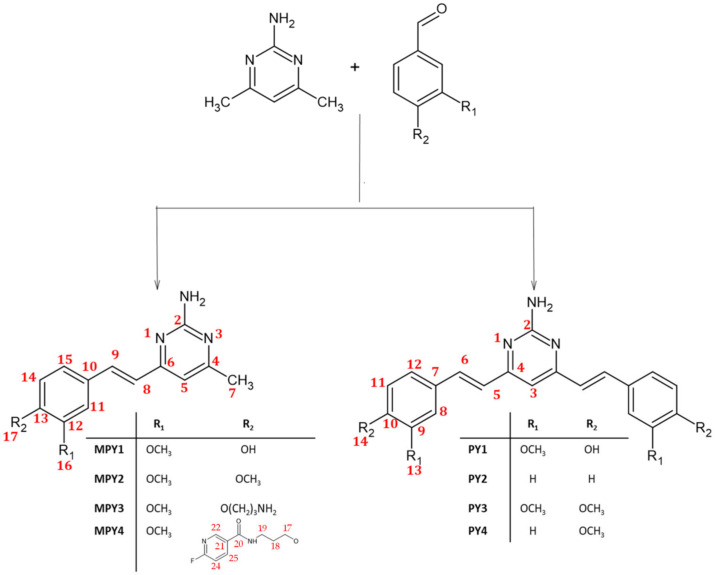
General scheme for the synthesis of amino-pyrimidine–curcumin derivatives, together with atom numbering used for NMR assignments. Red numbers refer to atom numbering used for NMR assignment.

**Figure 2 ijms-24-13963-f002:**
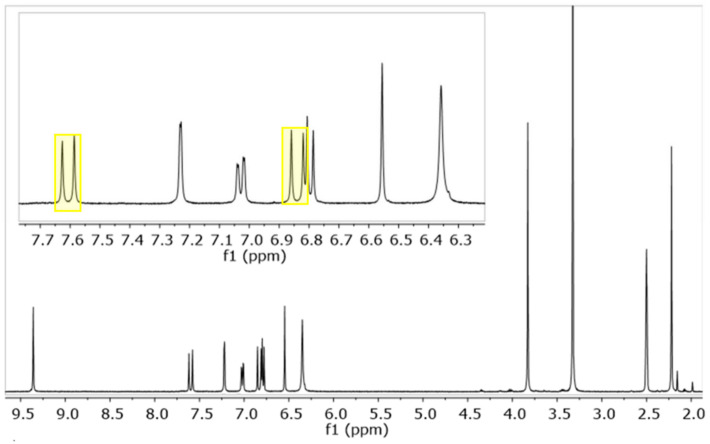
^1^H-NMR spectrum of MPY1 in DMSO-*d_6_* at 600 MHz (298 K). Highlighting boxes show the resonances of olefinic protons in the *E* configuration.

**Figure 3 ijms-24-13963-f003:**
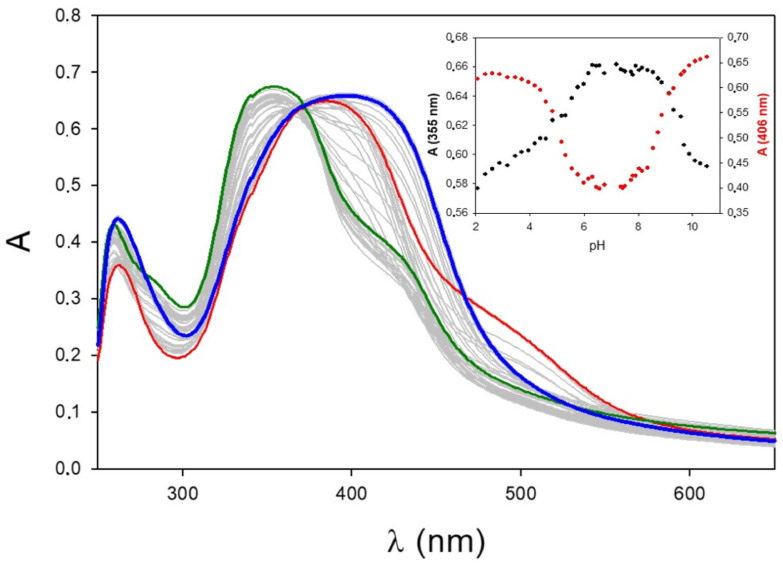
pH-metric spectrophotometric titration of MPY1 at 25 °C in aqueous solution ((MPY1) = 25 µM; (NaNO_3_) = 1 mM)) in the 250–650 nm spectral range. Red spectrum, pH = 2; green spectrum, pH = 7; blue spectrum, pH = 11. The inset shows the absorbance vs. pH at 355 nm (black) and 406 nm (red).

**Figure 4 ijms-24-13963-f004:**
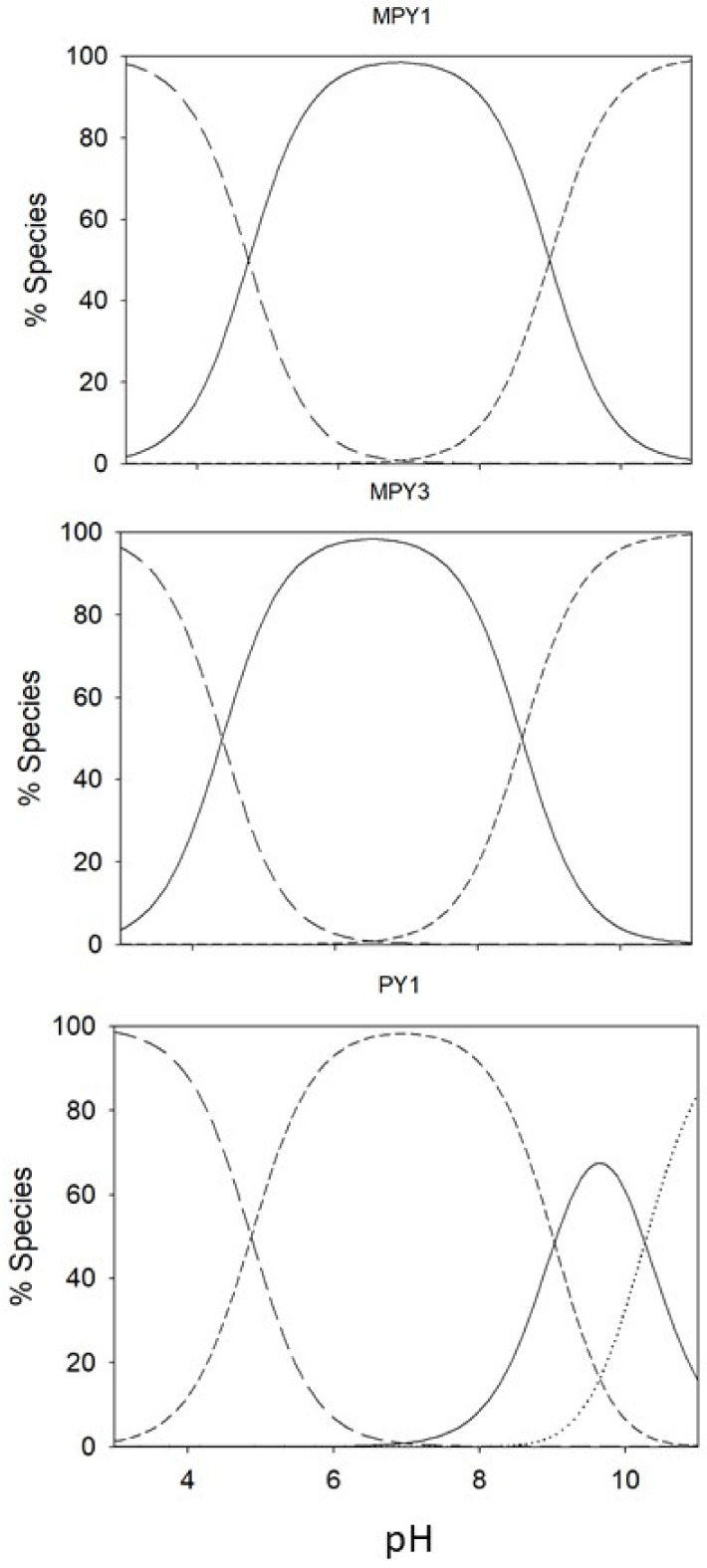
Species distribution curves for MPY1, MPY3, and PY1 (from top to bottom); [L]_tot_ = 1 × 10^−4^ mol L^−1^.

**Figure 5 ijms-24-13963-f005:**
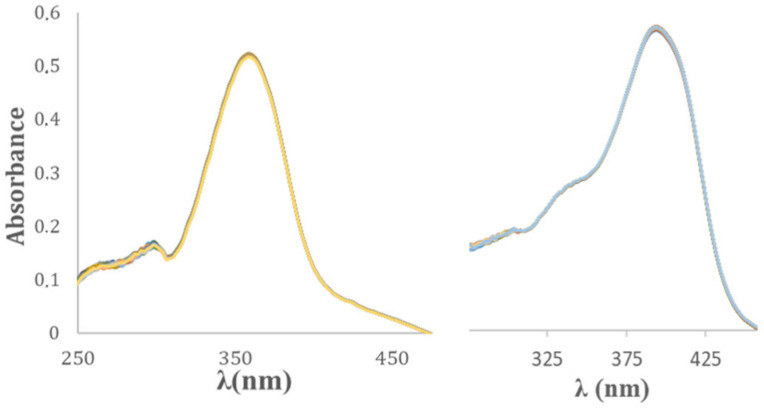
Absorption spectra of MPY1 (left) and PY1 (right) in PBS (phosphate-buffered saline) at 298 K ((MPY1) = (PY1) = 20 µM).

**Figure 6 ijms-24-13963-f006:**
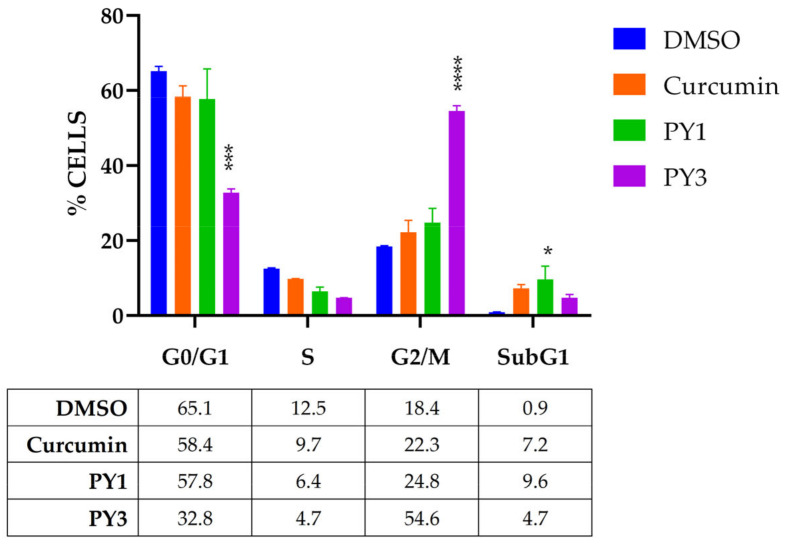
The histogram represents the distribution of LNCaP cells in the different phases of the cell cycle following the administration of DMSO, PY1, PY3, and curcumin for 48 h. The values represent means ± SEM of three independent experiments (two-way ANOVA with Fisher’s LSD test: * *p* < 0.05, *** *p* < 0.001, **** *p* < 0.0001, n = 3).

**Table 1 ijms-24-13963-t001:** Overall protonation constants (logβ_LH_) and pK_a_ values at 298 K; values are calculated from spectrophotometric data by HypSpec software [27]. Charges are omitted for clarity.

	MPY1 (H_2_L)	MPY3 (H_2_L)	PY1 (H_3_L)
logβ_11_	8.99 ± 0.01	8.62 ± 0.01	10.27 ± 0.03
p*K*_a3_	-	-	10.27 ± 0.03 ^c^
logβ_12_	13.72 ± 0.01	13.04 ± 0.01	19.30 ± 0.02
p*K*_a2_	8.99 ± 0.01 ^a^	8.62 ± 0.01 ^a^	9.04 ± 0.05 ^d^
logβ_13_	-	-	24.17 ± 0.05
p*K*_a1_	4.73 ± 0.02 ^b^	4.43 ± 0.02 ^b^	4.86 ± 0.07 ^e^

^a^ p*K*_a2_ = logβ_11_; ^b^ p*K*_a1_ = logβ_12_ − logβ_11_; ^c^ p*K*_a3_ = logβ_11_; ^d^ p*K*_a2_ = logβ_12_ − logβ_11_; ^e^ p*K*_a1_ = logβ_13_ − logβ_12_.

**Table 2 ijms-24-13963-t002:** GI50 values of the indicated compounds in PCa (LNCaP and PC3) and CRC (HT29 and HCT116) cell lines following 48 h treatments. n.a. = compounds not active at tested concentrations (≤100 µM). The values represent means ± SEM of three independent experiments.

GI50 (µM) ± SEM				
	LNCaP	PC3	HT29	HCT116
**MPY1**	73.9 ± 9.9	n.a.	n.a.	n.a.
**MPY2**	73.1 ± 14.0	84.4 ± 0.1	n.a.	n.a.
**MPY3**	68.8 ± 21.7	n.a.	n.a.	98.0 ± 13.6
**MPY4**	n.a.	n.a.	n.a.	n.a.
**PY1**	24.1 ± 8.5	23.5 ± 9.7	45.3 ± 4.1	19.9 ± 3.3
**PY2**	n.a.	n.a.		n.a.
**PY3**	12.1 ± 2.0	14.7 ± 0.9	19.7 ± 1.1	37.9 ± 11.6
**PY4**	n.a.	82.8 ± 23.4	n.a.	n.a.
**Curcumin**	35.8 ± 3.0	23.9 ± 0.4	30.6 ± 1.2	13.6 ± 2.9

## Data Availability

Not applicable.

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
