# Peer review of "Development of Stable Amino-Pyrimidine–Curcumin Analogs: Synthesis, Equilibria in Solution, and Potential Anti-Proliferative Activity"

_ijms, 2023, doi:10.3390/ijms241813963_

Round 1

Reviewer 1 Report

The current manuscript is an interesting study on the production of new curcumin derivatives for cancer treatment. Many relevant assays were performed, and the methodology is solid. Nevertheless, some alterations should be made before acceptance for publication:

- The authors should be more careful, since there are a few errors along the manuscript, for example: in the title page “Type of the Paper (Article)” should just say “Article”; entire website links should not appear in the text, as they do in line 36;

- An abbreviation list is missing;

- In the introduction section, more should be said to support why there is a need for new cancer therapies, namely current therapies and their limitations;

- “Scheme 1” should be called “Figure 1”; all figures should then be renumbered accordingly;

- Figure captions should include the meaning of the abbreviations that are used in the figures themselves;

- Figures 2 and 3 quality (resolution) should be improved;

- Bars in Figure 5 should be colored for better interpretation;

- Future studies should be discussed, namely the possible administration routes for the newly developed molecules, and formulation strategies, taking into account molecule stability and solubility (lipophilic/hydrophilic).

Author Response

Estimated Reviewer,

we have found all comments very pertinent and helpful to improve our article, in order to be presented at the best to the Readers of the IJMS.

All the changes are highlighted in yellow throughout text and here you can find the point-by-point reply to your comments and questions (our replies in blue).

Point-to-point replay to Reviewer 1 (R1)

(R1) The current manuscript is an interesting study on the production of new curcumin derivatives for cancer treatment. Many relevant assays were performed, and the methodology is solid.

(A) We thank the Reviewer for the interest in our work and the thoughtful comments.

Nevertheless, some alterations should be made before acceptance for publication:

- The authors should be more careful, since there are a few errors along the manuscript, for example: in the title page “Type of the Paper (Article)” should just say “Article”; entire website links should not appear in the text, as they do in line 36;

(A) The “Type of Article” has been corrected and the website has been added as a citation.

- An abbreviation list is missing;

(A) The abbreviation list has been added to facilitate the Reader.

- In the introduction section, more should be said to support why there is a need for new cancer therapies, namely current therapies and their limitations;

(A) The introduction was changed accordingly (p.1 and 2)

- “Scheme 1” should be called “Figure 1”; all figures should then be renumbered accordingly;

(A) Done

- Figure captions should include the meaning of the abbreviations that are used in the figures themselves;

(A) All the abbreviations included in the figures’ captions have been explicitly defined.

- Figures 2 and 3 quality (resolution) should be improved;

(A) Graphical quality of figures 2 and 3 was improved.

- Bars in Figure 5 should be colored for better interpretation;

(A) Figure 5 was changed accordingly.

- Future studies should be discussed, namely the possible administration routes for the newly developed molecules, and formulation strategies, taking into account molecule stability and solubility (lipophilic/hydrophilic).

(A) We have added a final statement at the end of the discussion section in relation to the potential oral administration of these new compounds.

Reviewer 2 Report

The manuscript Development of stable amino-pyrimidine curcumin analogs: synthesis, equilibria in solution and potential anti-proliferative activity is a nice attempt. The authors have synthesized eight new pyrimidine-curcumin derivatives, the synthesis was confirmed through HNMR and Absorbance spectra was also supports the work.

The claim of authors that prove that compounds were extremely stable in simulated physiological media (phosphate buffered saline and simulated plasma) is valid and the spectroscopic tools also prove this claim. The compounds were also tested in vitro to assess their effect on cell via- ability in prostate cancer ( LNCaP and PC3 ) and colorectal cancer (HT29 and HCT116) cell lines. Two  compounds the same anti-proliferative activity against HCT116 cells and improved cytotoxity against PC3 cell, the results of these activities are interesting and are beneficial for the readers, I recommend this manuscript for publication after the incorporation of following points:

1. English language needs attention, there are lot of spelling maistake/ syntax error, the author needs to check English throughout the manuscript

2.  First few lines of the abstract are non informative and useless "With the clear need for better cancer treatment, naturally occurring molecules represent a powerful inspiration. Recently, Curcumin has attracted attention for its pleiotropic anticancer activity in vitro, especially against colorectal and prostate cancer cells. Unfortunately, these encouraging results were disappointed in vivo, due to its low stability and poor bioavailability. To overcome these issues, analogs have been designed and synthesized" these lines should be removed.

3. The discussion portion is well written but this portion should be supported with the latest work

English language is moderate and needs correction through out

Author Response

Estimated Reviewer,

we have found all comments very pertinent and helpful to improve our article, in order to be presented at the best to the Readers of the IJMS.

All the changes are highlighted in yellow throughout text and here you can find the point-by-point reply to your comments and questions (our replies in blue).

Point-to-point replay to Reviewer 2 (R2)

The manuscript”Development of stable amino-pyrimidine curcumin analogs: synthesis, equilibria in solution and potential anti-proliferative activity” is a nice attempt. The authors have synthesized eight new pyrimidine-curcumin derivatives, the synthesis was confirmed through HNMR and Absorbance spectra was also supports the work.

The claim of authors that prove that compounds were extremely stable in simulated physiological media (phosphate buffered saline and simulated plasma) is valid and the spectroscopic tools also prove this claim. The compounds were also tested in vitro to assess their effect on cell viability in prostate cancer ( LNCaP and PC3 ) and colorectal cancer (HT29 and HCT116) cell lines. Two  compounds the same anti-proliferative activity against HCT116 cells and improved cytotoxity against PC3 cell, the results of these activities are interesting and are beneficial for the readers, I recommend this manuscript for publication after the incorporation of following points.

(A) We thank the Reviewer for the interest in our work and the thoughtful comments.

  1. English language needs attention, there are lot of spelling maistake/ syntax error, the author needs to check English throughout the manuscript.

(A) We apologize for the typos, the manuscript has been revised accordingly.

  1. First few lines of the abstract are non informative and useless "With the clear need for better cancer treatment, naturally occurring molecules represent a powerful inspiration. Recently, Curcumin has attracted attention for its pleiotropic anticancer activity in vitro, especially against colorectal and prostate cancer cells. Unfortunately, these encouraging results were disappointed in vivo, due to its low stability and poor bioavailability. To overcome these issues, analogs have been designed and synthesized" these lines should be removed.

(A) We concur with the Reviewer that the first abstract lines are not so informative, however we believe that they may help the Reader to better contextualize the manuscript and catch the attention. The first abstract lines were slightly reduced in the revised version of the manuscript.

  1. The discussion portion is well written but this portion should be supported with the latest work

(A) The discussion section was supported by the data that, to our knowledge, were recently published. If the Reviewer would like to suggest some more specific citations we could evaluate to add them.

Reviewer 3 Report

Please indicate in the abstract section how long the authors used the product they developed in their in vitro analysis against cancer cells.

Also, let him add the characterization methods of the product to the abstract section, which methods were used???

There is enough information for curcuma longa in the introduction section, but the literature data is missing, in vitro analysis result data should be compared in similar studies. The following article will help. Authors should be careful to specify these comparisons by duration.

Biocidal Activity of Bone Cements Containing Curcumin and Pegylated Quaternary Polyethylenimine

T Eren, G Baysal, F Dogan

Journal of Polymers and the Environment 28, 2469-2480

More literature is needed in the Introduction section.

-Please add the amounts and concentrations of the chemicals used in the synthesis steps,

-The authors did not specify the time in cell profiling analysis, please check

-Please refer to the literature on the formulas and equations used

Please explain why cell viability analyzes were applied to only 48 hours and why 24 hours were not observed first.

-please authors create a resistance table for cancer cell analysis and compare the results with the literature

Minor editing of English language required

Author Response

Estimated Reviewer,we have found all comments very pertinent and helpful to improve our article, in order to be presented at the best to the Readers of the IJMS.

All the changes are highlighted in yellow throughout text and here you can find the point-by-point reply to your comments and questions (our replies in blue).

Point-to-point replay to Reviewer 3 (R3)

Please indicate in the abstract section how long the authors used the product they developed in their in vitro analysis against cancer cells.

(A) The incubation treatment of 48 h was added in the abstract.

Also, let him add the characterization methods of the product to the abstract section, which methods were used???

(A) We thank the Reviewer for this suggestion, the abstract in the revised manuscript reports the analytical techniques used to chemically characterize the compounds.

There is enough information for curcuma longa in the introduction section, but the literature data is missing, in vitro analysis result data should be compared in similar studies. The following article will help. Authors should be careful to specify these comparisons by duration.

Biocidal Activity of Bone Cements Containing Curcumin and Pegylated Quaternary Polyethylenimine

T Eren, G Baysal, F Dogan

Journal of Polymers and the Environment 28, 2469-2480

More literature is needed in the Introduction section.

(A) The introduction has been enlarged and more references were added as suggested by the Reviewer.

-Please add the amounts and concentrations of the chemicals used in the synthesis steps,

(A) Done

-The authors did not specify the time in cell profiling analysis, please check

(A) 48h, added

-Please refer to the literature on the formulas and equations used

(A) Done

Please explain why cell viability analyzes were applied to only 48 hours and why 24 hours were not observed first.

(A) The 48h treatment was selected in order to have results comparable to previously reported studies, furthermore since the GI50 values are quite high after 48h treatment, we guess they should be even higher for shorter treatment.

-please authors create a resistance table for cancer cell analysis and compare the results with the literature

(A) In the discussion section, the results have been compared to those reported in the literature for similar compounds, particularly curcumin. Since the more promising results are only for two compounds (PY1 and PY3) we think an additional table is not necessary, since the overall results are summarized in table 2.